# Propofol-based total intravenous anesthesia is associated with less postoperative recurrence than desflurane anesthesia in thyroid cancer surgery

**Wei-Chieh Chiu**[1©], **Zhi-Fu Wu**[1,2,3,4©], **Meei-Shyuan Lee**[5], **Jamie Yu-Hsuan Chen**[5], **Yi-Hsuan Huang**[1], **Wei-Cheng Tseng**[1], **Hou-Chuan Lai**[1] *

1 Department of Anesthesiology, Tri-Service General Hospital and National Defense Medical Center, Taipei, Taiwan, Republic of China, 2 Department of Anesthesiology, Kaohsiung Medical University Hospital, Kaohsiung Medical University, Kaohsiung, Taiwan, Republic of China, 3 Department of Anesthesiology, Faculty of Medicine, College of Medicine, Kaohsiung Medical University, Kaohsiung, Taiwan, Republic of China, 4 Center for Regional Anesthesia and Pain Medicine, Wan Fang Hospital, Taipei Medical University, Taipei, Taiwan, ROC, 5 School of Public Health, National Defense Medical Center, Taipei, Taiwan, Republic of China

© These authors contributed equally to this work.

* m99ane@gmail.com

**Data Availability Statement:** All relevant data are within the paper and its Supporting Information files.

## Abstract

### Background

The effects of anesthesia in patients undergoing thyroid cancer surgery are still not known. We investigated the relationship between the type of anesthesia and patient outcomes following elective thyroid cancer surgery.

### Methods

This was a retrospective cohort study of patients who underwent elective surgical resection for papillary thyroid carcinoma between January 2009 and December 2019. Patients were grouped according to the type of anesthesia they received, desflurane or propofol. A Kaplan-Meier analysis was conducted, and survival/recurrence curves were presented from the date of surgery to death/recurrence. Univariable and multivariable Cox regression models were used to compare hazard ratios for recurrence after propensity matching.

### Results

A total of 621 patients (22 deaths, 3.5%) under desflurane anesthesia and 588 patients (32 deaths, 5.4%) under propofol anesthesia were included. Five hundred and eighty-eight patients remained in each group after propensity matching. Propofol anesthesia was not associated with better survival compared to desflurane anesthesia in the matched analysis (P = 0.086). However, propofol anesthesia was associated with less recurrence (hazard ratio, 0.38; 95% confidence interval, 0.25–0.56; P < 0.001) in the matched analysis.

**Funding:** The author(s) received no specific funding for this work.

**Competing interests:** The authors have declared that no competing interests exist.

**Abbreviations:** ASA, American Society of Anesthesiology; Ce, effect-site concentration; CCI, Charlson comorbidity index; CI, confidence interval; EtCO$_2$, end-tidal carbon dioxide; GA, general anesthesia; HIF-1α, hypoxia-inducible factor 1-alpha; HR, hazard ratio; IL, interleukin; INHA, inhalation anesthesia; IRB, institutional review board; METs, metabolic equivalents; NK, natural killer; PS, propensity score; PTC, papillary thyroid carcinoma; RCT, randomised controlled trial; SD, standard deviation; TCI, target-controlled infusion; TNF-α, tumor necrosis factor- alpha; TSGH, Tri-Service General Hospital.

## Conclusions

Propofol anesthesia was associated with less recurrence, but not mortality, following surgery for papillary thyroid carcinoma than desflurane anesthesia. Further prospective investigation is needed to examine the influence of propofol anesthesia on patient outcomes following thyroid cancer surgery.

## Introduction

Thyroid cancer is the most common malignant tumor in the endocrine system, accounting for about 1% of systemic malignancies, including papillary, follicular, un-differentiated, and medullary cancer [1, 2]. Papillary thyroid carcinoma (PTC) is the most common type of thyroid cancer, accounting for more than 80% of thyroid cancers and accompanied by lower malignancy and better prognosis [2]. However, PTC patients with invasion and metastases features may have a poor prognosis [2]. Although surgical resection plays an important role in the treatment of thyroid cancer, [1] surgical intervention may result in neuroendocrine and metabolic changes, which may impair cell-mediated immunity and activate the implantation of circulating tumor cells [3]. This potential combination of impaired immune responses and cancer cell seeding enhances the susceptibility of patients undergoing cancer surgery to the development of post-operative recurrence or metastasis associated with poor survival. Du et al. showed that sevoflurane-dexmedetomidine general anesthesia (GA) combined with cervical plexus nerve block could reduce the postoperative stress and inflammatory responses without mention of survival/recurrence in thyroid cancer patients [4]. The potential role of anesthetic techniques in cancer survival, postoperative recurrence, or metastasis formation has attracted attention [3].

Data from human cancer cell lines and animal research showed that different anesthetics might affect the immune system in different paths [5–11]. Inhalation anesthesia (INHA) have been shown to suppress natural killer (NK) cell cytotoxicity, attenuate neutrophil recruitment and phagocytosis, induce T-lymphocyte apoptosis, suppress release of interleukin (IL)-1β and tumor necrosis factor-α (TNF-α) from human peripheral mononuclear cells. Thus, INHA may promote immunosuppression and the metastatic spread of residual cancer cells postoperatively. In addition, INHA are associated with the upregulation of hypoxia-inducible factor 1-alpha (HIF-1α) in tumor cells, increasing transcription of genes encoding vascular endothelial growth factor, platelet-derived growth factor, and matrix metalloproteinases and thereby facilitating tumor angiogenesis, residual cell survival, and tumor cell migration [12, 13]. However, Wu et al. reported that isoflurane increases the concentration of proinflammatory cytokines in mouse brain tissue, a potentially protective effect for brain tumors [14]. Another research reported that sevoflurane inhibited migration and invasion, while enhancing cancer cell apoptosis [15]. Therefore, the effect of INHA on cancer progress is still inconclusive. By contrast, propofol has been shown to minimize perioperative immunosuppression by preserving NK cell and cytotoxic T cell function, inhibit macrophage function, against cancer cell dissemination and development of metastasis by regulating key cell signaling pathways implicated in tumorigenesis, such as the mitogen-activated protein kinase, nuclear factor kappa-B pathways, as well as regulating expression of miRNA and HIF-1α [13]. Therefore, propofol seemed to possess anti-cancer properties of reducing different tumor growth and decreasing the risk of recurrence [7, 11–13].

However, to date, rare studies have compared the effects of desflurane versus propofol anesthesia on patient outcomes following surgery for PTC. We hypothesized that patients under desflurane anesthesia might have subsequent poor outcomes than patients under propofol anesthesia, as in our previous cancer studies [16–24]. Thus, we conducted a retrospective cohort study to examine whether the choice of anesthesia, desflurane versus propofol, is associated with patient survival and postoperative recurrence after surgery for PTC.

## Materials and methods

This study was conducted at the Tri-Service General Hospital (TSGH), Taipei, Taiwan, Republic of China. The ethics committee of the TSGH approved this retrospective cohort study and waived the need for informed consent (TSGHIRB No: B202105146). The data was gathered from the electronic database and medical records of the TSGH. From January 2009 to December 2019, 1259 consecutive PTC patients with an American Society of Anesthesiologists (ASA) score of II–III who underwent elective surgery for primary PTC under propofol anesthesia (n = 588) or desflurane anesthesia (n = 621) were eligible for analysis. The type of anesthesia was chosen according to the anesthesiologist's personal preference. The exclusion criteria were propofol anesthesia combined with INHA, INHA other than desflurane, incomplete data, age < 20 years, thyroid cancer other than PTC; fifty cases were excluded (Fig 1). The data was accessed for research purposes since October 2021.

No medication was administrated before anesthesia induction. Each patient received standard monitoring, including electrocardiography (lead II), noninvasive blood pressure testing, pulse oximetry, and end-tidal carbon dioxide ($EtCO_2$) measurement. Anesthesia was induced by fentanyl, propofol, and cisatracurium (or rocuronium) in all patients [21].

As our previous reports [16–24], in brief, propofol anesthesia was maintained at an effect-site concentration (Ce) of 3.0–4.0 μg/mL by a target-controlled infusion (TCI) system (Fresenius Orchestra Primea; Fresenius Kabi AG, Bad Homburg, Germany); desflurane vaporizer was maintained between 4% and 10% (target minimum alveolar concentration of 0.7–1.3) [25]. During maintenance of anesthesia, all patients received FiO2 of 50–100% oxygen at a flow rate of 0.3–1.0 L/min in a closed breathing system, and desflurane or Ce of propofol was adjusted downward and upward by desflurane 0.5–2.0% or propofol Ce 0.2–0.5 μg/mL, respectively, if needed based on hemodynamics. Repetitive bolus injections of fentanyl and cisatracurium (or rocuronium) were administered as necessary during surgery. The level of $EtCO_2$ was maintained at 35–45 mmHg [16–24]. All patients received complete surgical resection as possible and were extubated, then transferred to the post-anesthesia care unit after surgery.

### Variables

We retrospectively gathered the following patient data: the type of anesthesia; calendar period; sex; age at the time of surgery. We used the Charlson Comorbidity Index (CCI) to predict 10-year survival in patients with multiple comorbidities [21]. Preoperative functional status was assessed in metabolic equivalents (METs). As cardiac and long-term risks increase in patients with a functional capacity of < 4 METs during activities of daily living [26], patients were grouped according to whether the value was ≥ 4 METs or < 4 METs [21]. We also used the Clavien-Dindo classification, scaled from 0 (no complication) to V (most complications), to grade surgical complications. Other data included ASA physical status scores (ranging from I, indicating lowest morbidity, to V, indicating highest morbidity); tumor size; tumor number; surgical procedure; pathological staging (p-TNM); poor differentiation of the primary tumor; positive margin of the primary tumor; postoperative radiation therapy; postoperative chemotherapy; postoperative hormone (levothyroxine) therapy; postoperative iodine-131 (I-131)

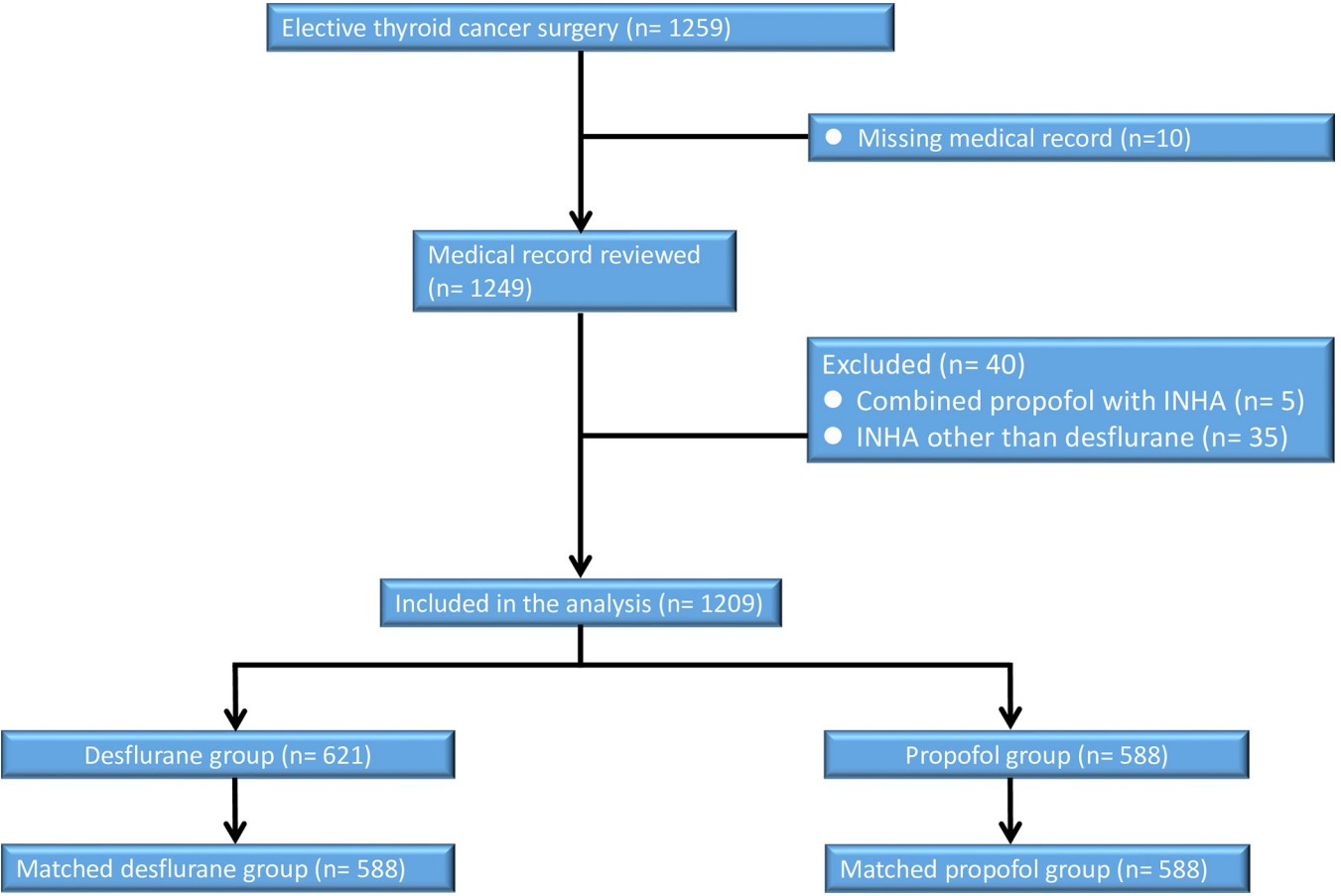

**Fig 1. Flow diagram detailing the selection of patients included in the retrospective analysis.** 50 patients were excluded due to combined propofol anesthesia with inhalation anesthesia (INHA), INHA other than desflurane, incomplete data, age < 20 years, and thyroid cancer other than papillary thyroid carcinoma.

treatment; secondary surgery; the presence of postoperative recurrence; the presence of postoperative metastasis. Because these variables have been shown or posited to affect patient outcomes, they were chosen as potential confounders [21].

## Statistical methods

The main outcomes were overall survival and postoperative recurrence, which were compared between the propofol and desflurane anesthesia. The survival/recurrence time was defined as the interval between the date of surgery and the date of death/recurrence on September 14, 2022, for those who were censored. We collected the data of the recurrence by the clinical data support of the Cancer Registry Group of TSGH based on patients who developed any evidence of recurrence (such as abnormal sonographic and cytology findings, thyroglobulin levels, and iodine uptake beyond the thyroid bed in a diagnostic whole-body scan) after achieving an excellent response to the primary surgery [27]. All data are shown as mean ± standard deviation (SD) or number (percentage) [21].

Mortality rates and patient characteristics were compared between the groups treated with the different anesthetics using Student's t test or the chi-square test. The survival based on the type of anesthesia was depicted visually in a Kaplan-Meier survival curve. The association between the type of anesthesia (propofol or desflurane) and postoperative recurrence was

analyzed by the Cox proportional-hazards model with and without adjustment for the above-mentioned variables [21]. To avoid multicollinearity, if there is a high correlation between the independent variables, it will be excluded in the multivariable analysis. In addition, surgeon volume may affect prognosis of thyroid cancer after surgery [28]. Therefore, hazard ratios (HRs) were also adjusted by surgeons (n = 11) in the multivariable analysis.

The propensity scores (PS) were created by simple logistic regression model in order to deal with the differences between propofol and desflurane groups. The model was build based on the abovementioned variables. We obtained 588 matched pairs based on one-to-one matching, using SPSS Statistics Version 29.0 (IBM Corp., Armonk, NY, USA) with calipers at 0.2 SD of the logit of the propensity score and without replacement. Propofol or desflurane anesthesia in a 1:1 ratio, to make sure the comparability between propofol and desflurane anesthesia before the surgery. Two-tailed P-values less than 0.05 were considered statistically significant.

## Results

The patients' and treatment characteristics are shown in Table 1. There were significant differences in calendar periods, surgical procedures, positive margins, grade of surgical complications, postoperative hormone and I-131 treatment, and secondary surgery between the two anesthetic techniques. Age, CCI, preoperative functional status, ASA score, tumor size, tumor number, p-TNM, poor differentiation, use of postoperative radiotherapy, and use of postoperative chemotherapy showed insignificant differences between the two anesthetic techniques (Table 1).

The overall postoperative recurrence rate was significantly lower in the propofol anesthesia group (5.8%) than in the desflurane anesthesia group (14.0%) during follow-up (P < 0.001). However, the overall mortality rate (propofol: 5.4% vs desflurane: 3.5%; P = 0.145) or the cancer-specific mortality rate (propofol: 5.4% vs desflurane: 3.5%; P = 0.145) did not differ between the two groups (Table 1). The mean follow-up time was 6.2 ± 3.2 years for the propofol group and 5.4 ± 3.3 years for the desflurane group. Overall survival curves for the two anesthetic techniques are shown in Fig 2A.

The overall recurrence risk associated with propofol and desflurane anesthesia after surgery for PTC is reported in Table 2. Overall recurrence from the date of surgery grouped according to the anesthetic technique and other variables was compared individually in a univariable Cox model and subsequently in a multivariable Cox regression model. Variables that significantly increased the recurrence risk were desflurane anesthesia, later calendar period (2018–2019), subtotal thyroidectomy, and p-TNM stage II after multivariable analysis (Table 2). Functional status was excluded from the model because it was the inverse of ASA scores. Patients under propofol anesthesia showed less overall recurrence than those under desflurane anesthesia, the crude HR was 0.39 (95% confidence interval (CI), 0.26–0.57; P < 0.001). This finding did not change substantially in the multivariable analysis (HR, 0.48; 95% CI, 0.30–0.76; P = 0.002) (Table 2). Overall recurrence curves for the two anesthetic techniques are shown in Fig 2B.

We used the PS from the logistic regression to adjust baseline characteristics and choice of therapy between the two anesthetic techniques due to significant differences in baseline characteristics between the two anesthetic techniques. Five hundred and eighty-eight pairs were formed after matching (Table 1). Patient characteristics and prognostic factors of primary PTC showed insignificant differences between matched groups (except calendar period and surgical procedure; Table 1). PS-matched survival/recurrence curves for the two anesthetic techniques are shown in Fig 2C and 2D. Propofol anesthesia was not associated with less mortality compared to desflurane anesthesia in the PS-matched analysis (propofol: 5.4% vs

**Table 1. Patients' and treatment characteristics and clinical outcomes for overall group and matched group after propensity scoring.**

| Variables | Overall Patients | | | Matched Patients | | | |
|---|---|---|---|---|---|---|---|
| | Propofol (n = 588) | Desflurane (n = 621) | p value | Propofol (n = 588) | Desflurane (n = 588) | p value | SMD |
| Calendar period, n (%) | | | <0.001 | | | <0.001 | 0.351 |
| 2009–2011 | 144 (25) | 119 (19) | | 144 (25) | 111 (19) | | |
| 2012–2014 | 183 (31) | 119 (19) | | 183 (31) | 105 (18) | | |
| 2015–2017 | 137 (23) | 170 (27) | | 137 (23) | 166 (28) | | |
| 2018–2019 | 124 (21) | 213 (34) | | 124 (21) | 206 (35) | | |
| Male sex, n (%) | 148 (25) | 167 (27) | 0.538 | 148 (25) | 158 (27) | 0.550 | 0.039 |
| Age (years), Mean (SD) | 46 (13) | 46 (14) | 0.894 | 46 (13) | 46 (14) | 0.893 | 0.008 |
| Charlson comorbidity index, Mean (SD) | 3.1 (2.0) | 3.1 (1.9) | 0.993 | 3.1 (2.0) | 3.1 (2.0) | 0.976 | 0.000 |
| Functional status, n (%) | | | 0.379 | | | 0.354 | 0.059 |
| < 4 MET | 88 (15) | 81 (13) | | 88 (15) | 76 (13) | | |
| ≥ 4 MET | 500 (85) | 540 (87) | | 500 (85) | 512 (87) | | |
| ASA, n (%) | | | 0.379 | | | 0.354 | 0.059 |
| II | 500 (85) | 540 (87) | | 500 (85) | 512 (87) | | |
| III | 88 (15) | 81 (13) | | 88 (15) | 76 (13) | | |
| Tumor size, Mean (SD) | 1.8 (0.9) | 1.8 (1.0) | 0.544 | 1.8 (1.0) | 1.8 (1.0) | 0.612 | 0.029 |
| Tumor number, n (%) | | | 0.826 | | | 0.804 | 0.010 |
| 1 | 455 (77) | 485 (78) | | 455 (77) | 456 (78) | | |
| 2 | 110 (19) | 109 (18) | | 110 (19) | 105 (18) | | |
| 3 | 23 (3.9) | 27 (4.3) | | 23 (3.9) | 27 (4.6) | | |
| Surgical procedure | | | 0.014 | | | 0.008 | 0.183 |
| Total thyroidectomy | 470 (80) | 456 (73) | | 470 (80) | 427 (73) | | |
| Subtotal thyroidectomy | 55 (9.4) | 65 (11) | | 55 (9.4) | 65 (11) | | |
| Lobectomy | 63 (11) | 100 (16) | | 63 (11) | 96 (16) | | |
| pTNM, n (%) | | | 0.684 | | | 0.472 | N/A |
| 1 | 395 (67) | 437 (70) | | 395 (67) | 417 (71) | | |
| 2 | 43 (7.3) | 42 (6.8) | | 43 (7.3) | 42 (7.1) | | |
| 3 | 126 (21) | 118 (19) | | 126 (21) | 105 (18) | | |
| 4 | 24 (4.1) | 24 (3.9) | | 24 (4.1) | 24 (4.1) | | |
| Poor dedifferentiation, n (%) | 27 (4.6) | 18 (2.9) | 0.161 | 27 (4.6) | 18 (3.1) | 0.171 | N/A |
| Positive margin, n (%) | 44 (7.5) | 68 (11) | 0.048 | 44 (7.5) | 66 (11) | 0.028 | N/A |
| Grade of surgical complications, n (%) | | | <0.001 | | | <0.001 | N/A |
| 0 | 579 (99) | 571 (92) | | 579 (99) | 540 (92) | | |
| I+II | 9 (1.5) | 50 (8.1) | | 9 (1.5) | 48 (8.2) | | |
| Radiotherapy, n (%) | 455 (77) | 475 (77) | 0.765 | 455 (77) | 449 (76) | 0.678 | N/A |
| Chemotherapy, n (%) | 3 (0.5) | 8 (1.3) | 0.262 | 3 (0.5) | 8 (1.4) | 0.226 | N/A |
| Hormone therapy, n (%) | 576 (98) | 580 (93) | <0.001 | 576 (98) | 548 (93) | <0.001 | N/A |
| I-131, n (%) | 497 (85) | 488 (79) | 0.008 | 497 (85) | 460 (78) | 0.005 | N/A |
| Secondary surgery, n (%) | 44 (7.5) | 71 (11) | 0.025 | 44 (7.5) | 70 (12) | 0.010 | N/A |
| Postoperative recurrence, n (%) | 34 (5.8) | 86 (14) | <0.001 | 34 (5.8) | 83 (14) | <0.001 | N/A |
| Postoperative metastasis, n (%) | 49 (8.3) | 50 (8.1) | 0.941 | 49 (8.3) | 47 (8.0) | 0.915 | N/A |
| All-cause mortality, n (%) | 32 (5.4) | 22 (3.5) | 0.145 | 32 (5.4) | 19 (3.2) | 0.086 | N/A |
| Cancer mortality, n (%) | 32 (5.4) | 22 (3.5) | 0.145 | 32 (5.4) | 19 (3.2) | 0.086 | N/A |

Propensity score matching only included those variables known at pre-operation.

Data shown as mean ± SD or n (%). MET = metabolic equivalents; ASA = American Society of Anesthesiologists; pTNM = pathological tumor–node–metastasis; Grade of surgical complications: Clavien-Dindo classification; N/A = not applicable because these variables were not included in propensity score matching.

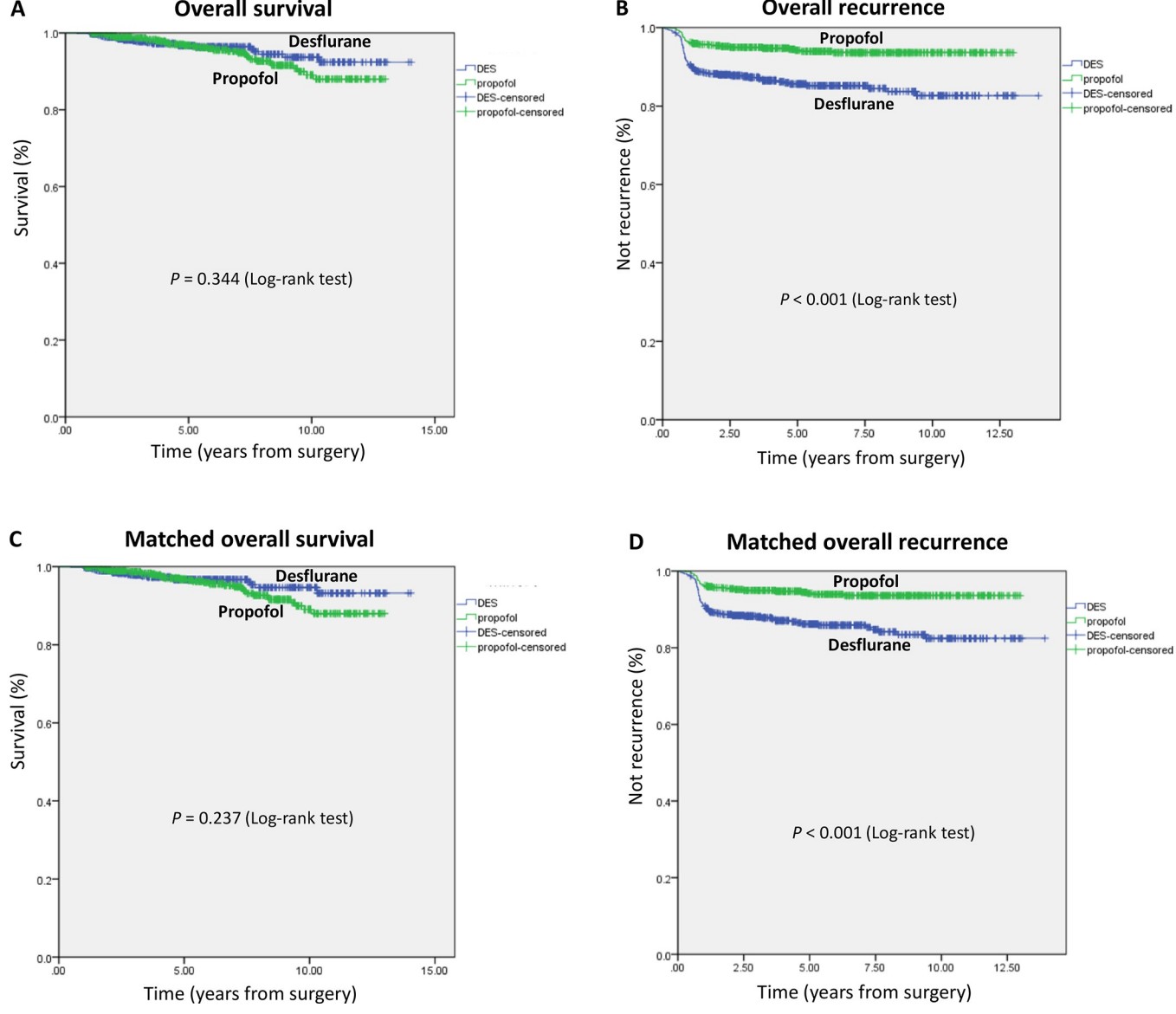

**Fig 2.** (A) Overall survival curves from the date of surgery by anesthesia type. (B) Overall recurrence curves from the date of surgery by anesthesia type. (C) Overall survival curves from the date of surgery by anesthesia type after propensity score matching. (D) Overall recurrence curves from the date of surgery by anesthesia type after propensity score matching.

desflurane: 3.2%; P = 0.086). However, propofol anesthesia was associated with less postoperative recurrence (propofol: 5.8% vs desflurane: 14.0%; P < 0.001) in the PS-matched analysis (Table 1).

## Risk of postoperative recurrence/mortality by the type of anesthesia

Patients with propofol anesthesia had less postoperative recurrence than those with desflurane anesthesia; the crude HR was 0.39 (95% CI, 0.26–0.57; P < 0.001); the crude HR with adjustment by calendar periods and surgeons was 0.49 (95% CI, 0.31–0.76; P = 0.002); the fully [variables were with a P-value less than 0.1 in the univariable analyses and surgeons (n = 11) without functional status (due to functional status was the reciprocal of ASA score) in the

**Table 2. Cox proportional hazards regression for recurrence: Univariable and multivariable models for overall patients.**

| | Univariable | | Multivariable | |
|---|---|---|---|---|
| Variables | HR (95% CI) | p value | HR (95% CI) | p value |
| Anesthesia, Propofol (ref: Desflurane) | 0.39 (0.26–0.57) | < 0.001 | 0.48 (0.30–0.76) | 0.002 |
| Calendar period (year; ref: 2009–2011) | | < 0.001 | | < 0.001 |
| 2012–2014 | 1.20 (0.52–2.01) | 0.962 | 1.30 (0.64–2.64) | 0.463 |
| 2015–2017 | 1.50 (0.77–2.92) | 0.239 | 1.51 (0.74–3.08) | 0.264 |
| 2018–2019 | 5.39 (2.98–9.76) | < 0.001 | 7.03 (3.52–14.0) | < 0.001 |
| Female (ref: Male) | 0.85 (0.57–1.26) | 0.406 | | |
| Age (years) | 1.01 (0.99–1.02) | 0.257 | | |
| Charlson comorbidity index | 1.08 (1.00–1.17) | 0.057 | 1.08 (0.96–1.23) | 0.215 |
| ASA III, (ref: II) | 1.22 (0.75–1.97) | 0.419 | | |
| Surgical procedure (ref: total thyroidectomy) | | <0.001 | | <0.001 |
| Subtotal thyroidectomy | 6.85 (4.65–10.1) | <0.001 | 6.15 (3.93–9.64) | <0.001 |
| Lobectomy | 1.75 (1.00–3.04) | 0.049 | 1.15 (0.64–2.08) | 0.635 |
| Tumor size (cm) | 1.05 (0.88–1.26) | 0.602 | | |
| Tumor number >1, (ref: 1) | 1.07 (0.77–1.48) | 0.699 | | |
| pTNM stage of primary tumor, (ref: I) | | < 0.001 | | 0.010 |
| II | 3.66 (2.28–5.87) | < 0.001 | 2.47 (1.44–4.22) | 0.001 |
| III | 1.09 (0.68–1.76) | 0.709 | 1.55 (0.80–3.01) | 0.198 |
| IV | 1.53 (0.66–3.52) | 0.321 | 1.12 (0.35–3.56) | 0.848 |
| Poor dedifferentiation (ref: no) | 1.97 (0.96–4.04) | 0.064 | 0.64 (0.24–1.71) | 0.372 |
| Positive margin (ref: no) | 3.09 (2.02–4.72) | <0.001 | 1.19 (0.71–1.98) | 0.516 |
| Grade of surgical complications, (ref: 0) | | | | |
| I+II | 1.37 (0.67–2.81) | 0.387 | | |
| Postoperative radiotherapy (ref: no) | 1.03 (0.66–1.58) | 0.913 | | |
| Postoperative chemotherapy (ref: no) | 3.31 (1.05–10.4) | 0.041 | 0.91 (0.24–3.51) | 0.889 |
| Hormone therapy | 0.95 (0.39–2.32) | 0.905 | | |
| I-131 (ref: no) | 1.03 (0.64–1.66) | 0.913 | | |

Sample sizes for propofol and desflurane groups were 588 and 621, respectively.

Adjusted-HRs were adjusted by those variables were with a p-value less than 0.1 in the univariable analyses and surgeons (n = 11). Functional status was excluded from the multivariable due to it was the reciprocal of ASA.

ASA = American Society of Anesthesiologists; pTNM = pathological tumor–node–metastasis.

Table 2)] adjusted crude HR was 0.48 (95% CI, 0.30–0.76; P = 0.002); the PS-matched HR was 0.38 (95% CI, 0.25–0.56; P < 0.001); and the PS-matched HR with adjustment by calendar periods and surgeons was 0.49 (95% CI, 0.31–0.77; P = 0.002); the fully adjusted PS-matched HR was 0.49 (95% CI, 0.31–0.78; P = 0.003; Table 3). However, propofol anesthesia was not associated with less postoperative mortality compared with desflurane anesthesia (Table 3).

Because sex, postoperative hormone therapy, and surgical procedure may affect postoperative recurrence [29, 30], subgroup analyses were stratified by these three variables. Patients who received propofol exhibited less postoperative recurrence than those who received desflurane, regardless of whether sex was male or female, and whether surgical procedure was total or subtotal thyroidectomy.

For male and female patients, the crude HRs were 0.32 (95% CI, 0.14–0.70; P = 0.004) and 0.42 (95% CI, 0.26–0.66; P < 0.001); the PS-matched HRs were 0.31 (95% CI, 0.14–0.69; P = 0.004) and 0.41 (95% CI, 0.26–0.65; P < 0.001), respectively. For female patients, the crude and PS-matched HRs with adjustment by calendar periods and surgeons were 0.49 (95% CI,

**Table 3. Hazards ratio (95% CI) (Propofol vs. Desflurane) by cox proportional-hazards regression for mortality/recurrence: Comparisons of various models and subgroup analyses.**

| Stratified variables | Overall Patients | | | PS-Matched Patients (588 pairs) | | |
|---|---|---|---|---|---|---|
| | Crude HR (95% CI)/ P value | Adjusted HR by calendar period and surgeons (95% CI)/ P value | Fully adjusted HR (95% CI)/ P value | PS-matched-HR (95% CI)/ P value | Adjusted HR by calendar period and surgeons (95% CI)/ P value | Fully adjusted HR (95% CI)/ P value |
| *Mortality* | 1.30 (0.75–2.24)/ | 1.94 (0.93–4.06)/ | 1.50 (0.70–3.25)/ | 1.41 (0.80–2.49)/ | 2.31 (1.04–5.13)/ | 1.73 (0.75–3.85)/ |
| | 0.346 | 0.079 | 0.300 | 0.239 | 0.039 | 0.203 |
| *Recurrence* | | | | | | |
| **Non-stratified** | 0.39 (0.26–0.57)/ | 0.49 (0.31–0.76)/ | 0.48 (0.30–0.76)/ | 0.38 (0.25–0.56)/ | 0.49 (0.31–0.77)/ | 0.49 (0.31–0.78)/ |
| | <0.001 | 0.002 | 0.002 | <0.001 | 0.002 | 0.003 |
| **Sex** | | | | | | |
| Male | 0.32 (0.14–0.70)/ | cannot converge/ | cannot converge/ | 0.31 (0.14–0.69)/ | cannot converge/ | cannot converge/ |
| | 0.004 | - | - | 0.004 | - | - |
| Female | 0.42 (0.26–0.66)/ | 0.49 (0.29–0.83)/ | 0.41 (0.24–0.70)/ | 0.41 (0.26–0.65)/ | 0.50 (0.30–0.85)/ | 0.43 (0.25–0.74)/ |
| | <0.001 | 0.007 | 0.001 | <0.001 | 0.010 | 0.002 |
| **Hormone therapy** | | | | | | |
| Yes | 0.36 (0.24–0.54)/ | 0.47 (0.30–0.74)/ | 0.46 (0.28–0.73)/ | 0.35 (0.23–0.53)/ | 0.47 (0.30–0.75)/ | 0.47 (0.29–0.75)/ |
| | <0.001 | 0.001 | 0.001 | <0.001 | 0.002 | 0.002 |
| No | 2.49 (0.42–14.9)/ | cannot converge/ | cannot converge/ | 2.42 (0.41–14.5)/ | cannot converge/ | cannot converge/ |
| | 0.319 | - | - | 0.332 | - | - |
| **Surgical procedure** | | | | | | |
| Total thyroidectomy | 0.27 (0.14–0.49)/ | 0.36 (0.18–0.73)/ | 0.31 (0.15–0.65)/ | 0.26 (0.14–0.49)/ | 0.37 (0.18–0.74)/ | 0.32 (0.15–0.68)/ |
| | <0.001 | 0.005 | 0.002 | <0.001 | 0.005 | 0.003 |
| Subtotal thyroidectomy | 0.48 (0.26–0.89)/ | 0.43 (0.21–0.88)/ | 0.58 (0.27–1.22)/ | 0.48 (0.26–0.89)/ | 0.43 (0.21–0.88)/ | 0.58 (0.27–1.22)/ |
| | 0.019 | 0.020 | 0.150 | 0.019 | 0.020 | 0.150 |
| Lobectomy | 0.95 (0.35–2.61)/ | cannot converge/ | cannot converge/ | 1.01 (0.36–2.85)/ | cannot converge/ | cannot converge/ |
| | 0.920 | - | - | 0.979 | - | - |

Fully adjusted-HRs were adjusted by those variables were with a p-value less than 0.1 in the univariable analyses and surgeons (n = 11) without functional status (due to functional status was the reciprocal of ASA) in the Table 2. HR = hazard ratio; PS = propensity score.

0.29–0.83; P = 0.007) and 0.50 (95% CI, 0.30–0.85; P = 0.010); the fully adjusted crude and PS-matched HRs were 0.41 (95% CI, 0.24–0.70; P = 0.001) and 0.43 (95% CI, 0.25–0.74; P = 0.002), respectively (Table 3).

For total thyroidectomy and subtotal thyroidectomy, the crude HRs were 0.27 (95% CI, 0.14–0.49; P < 0.001) and 0.48 (95% CI, 0.26–0.89; P = 0.019); the crude HRs with adjustment by calendar periods and surgeons were 0.36 (95% CI, 0.18–0.73; P = 0.005) and 0.43 (95% CI, 0.21–0.88; P = 0.020); the PS-matched HRs were 0.26 (95% CI, 0.14–0.49; P < 0.001) and 0.48 (95% CI, 0.26–0.89; P = 0.019); and the PS-matched HRs with adjustment by calendar periods and surgeons were 0.37 (95% CI, 0.18–0.74; P = 0.005) and 0.43 (95% CI, 0.21–0.88; P = 0.020, respectively. Meanwhile, for total thyroidectomy, the fully adjusted crude and PS-matched HRs were 0.31 (95% CI, 0.15–0.65; P = 0.002) and 0.32 (95% CI, 0.15–0.68; P = 0.003; Table 3).

In addition, for use of postoperative hormone therapy, the crude and PS-matched HRs were 0.36 (95% CI, 0.24–0.54; P < 0.001) and 0.35 (95% CI, 0.23–0.53; P < 0.001); the crude and PS-matched HRs with adjustment by calendar periods and surgeons were 0.47 (95% CI, 0.30–0.74; P = 0.001) and 0.47 (95% CI, 0.30–0.75; P = 0.002); the fully adjusted crude and PS-matched HRs were 0.46 (95% CI, 0.28–0.73; P = 0.001) and 0.47 (95% CI, 0.29–0.75; P = 0.002), respectively (Table 3).

In summary, patients under desflurane anesthesia had higher postoperative recurrence than those under propofol anesthesia with or without adjustment by calendar periods and surgeons following surgery for PTC. Propofol anesthesia was associated with less postoperative recurrence regardless of whether sex was male or female, and whether surgical procedure was total or subtotal thyroidectomy, and use of postoperative hormone therapy. However, there was no significant difference in mortality between the two groups. Finally, there was no occurrence of cardiovascular or adverse events in the two groups perioperatively.

## Discussion

In the literature, we first report that propofol anesthesia reduced 52% recurrence rate compared with desflurane anesthesia following surgical resection for PTC. However, there is no significant difference between the two groups in overall survival or cancer-specific survival. Our results suggest a potential effect in humans, and it seems biologically implausible that something as complicated as cancer can be reduced by more than a factor of two simply by anesthetic selection. In addition, our results most likely overestimate the true treatment effect, which is common in retrospective studies. Until now, there are rare studies on the influence of anesthetic techniques in PTC patients; further large randomized controlled trials are needed to examine the role of anesthetic techniques on postoperative outcome in surgery for PTC.

The 5- year mortality following PTC surgery was about 3.2% and the recurrence ranges from 5% to 21% [31]. In addition, recurrence with metastatic lymph nodes might decrease 5-year survival [32]. Our results were consistent with the abovementioned study [31]. However, propofol anesthesia was associated with lower recurrence rate, but not metastasis or mortality rate in this study. The conflict results might result from that most of patients with postoperative recurrence received I-131 therapy, and it seemed to decrease postoperative metastasis [33]. Further investigations are needed.

Surgical resection is the gold standard of therapy for PTC; however, surgery may suppress important host defenses and stimulate the development of recurrence. Postoperative recurrence has an impact on patient prognosis and survival in PTC. Thus, research on thyroid cancer has focused on developing strategies to ameliorate overall patient survival via reducing postoperative recurrence [32]. The plausibility of tumor recurrence depends on the balance between the cancer invasive potential and the host defense, of which NK cell function and cell-mediated immunity are important parts [34, 35]. Data from studies on human cancer cell lines and animal showed that different anesthetic techniques or anesthetics could influence immune response [5–10]. and may affect risks of cancer recurrence, metastasis, or patient survival [7, 9–11].

Data from human thyroid cell lines support the influence of propofol on thyroid cancer cell growth and survival via different pathways [2, 36, 37]. Li et al. reported that propofol suppressed migration, invasion, and epithelial-mesenchymal transition in PTC cells by regulating miR-122 expression [2]. Li et al. reported that propofol upregulated miR-320a and reduced HMGB1 by downregulating ANRIL to inhibit PTC cell malignant behaviors [36]. Zhang et al. showed that propofol inhibited thyroid cancer cell (but not PTC) proliferation, migration, and invasion by suppressing SHH and PI3K/AKT signaling pathways via the miR-141-3p/BRD4 axis [37]. In addition, Li et al. showed that sevoflurane inhibited migration and invasion, while

enhanced cell apoptosis by downregulating miR-155 in PTC cells [15]. However, in the literature, there is no study on the effect of desflurane on thyroid cancer cells. Thus, our results show that propofol may reduce the risk of recurrence by suppressing PTC growth, whereas desflurane may cause opposite effects on PTC growth. Further investigations are needed to recheck this association.

This study also found that a later calendar period (2018–2019), subtotal thyroidectomy and higher p-TNM stage were associated with higher recurrence rate after surgery for PTC. The higher recurrence rate of the later calendar period (2018–2019) might result from some anesthesiologists who prefer propofol anesthesia leaving from our hospital (124 cases with propofol anesthesia and 213 cases with desflurane anesthesia). Kuo and Wang [29] reported that subtotal thyroidectomy was one of the PTC recurrence risk factors, which was compatible with our results. In addition, higher p-TNM stages were associated with higher recurrence rates, which was similar with previous study [38]. Further investigation is still necessary.

There were some limitations in this study. First, it was retrospective, and the 1209 patients were not randomly allocated. However, we used all available patients from January 2009 to December 2019 from the medical center. Patient characteristics such as calendar period differed significantly between the groups, and we conducted PS matching to address this issue. PS matching including those variables which information could be known before operation to mimic RCT. The variables were calendar period, sex, age, CCI, functional status, ASA score, tumor size, tumor number, and surgical procedure, except surgeon volume due to it cannot converge in the estimation. Kim et al. reported that high-volume surgeons were significantly associated with less recurrence, but not distant metastasis or cancer-specific death in patients with PTC [28]. Since the standardized mean difference for calendar period and surgical procedure was greater than 0.1, these variables, as well as surgeon volume, were included in the modeling (Table 3). Finally, the findings did not change substantially using further adjustment by calendar period, surgical procedure and surgeons (Table 3). Second, we only analyzed PTC because it is the most common type of thyroid cancer [2]. Third, different INHA may have varying effects on thyroid cancer. This study focused on desflurane because it is the most frequently used INHA in our hospital. Fourth, patients maintained with desflurane also received single bolus 1–2 mg/kg propofol for induction of anesthesia, and its effect on our findings is unknown. However, Schaefer et al. reported that the increasing doses of propofol (per 10 mg/kg) did not associate with decreased one-year mortality in patients with solid tumors [39]. Fifth, the low mortality rate in this study, limited by the characteristics of thyroid cancer, may lead to lower statistical efficacy.

## Conclusions

Propofol anesthesia was associated with less postoperative recurrence than desflurane anesthesia following surgery for PTC. However, propofol anesthesia was not associated with better survival compared to desflurane anesthesia in surgery for PTC.

## Supporting information

**S1 Table. Surgeon volume (n = 11) in multivariable models for overall patients (Cox proportional hazards regression for recurrence).**
(DOCX)

**S1 Data. minimal data set.**
(XLSX)

## Acknowledgments

The authors thank the Cancer Registry Group of Tri-Service General Hospital for the clinical data support.

## Author Contributions

**Conceptualization:** Zhi-Fu Wu, Hou-Chuan Lai.

**Data curation:** Wei-Chieh Chiu, Yi-Hsuan Huang, Wei-Cheng Tseng.

**Formal analysis:** Meei-Shyuan Lee, Jamie Yu-Hsuan Chen.

**Investigation:** Wei-Chieh Chiu, Yi-Hsuan Huang, Hou-Chuan Lai.

**Methodology:** Zhi-Fu Wu, Meei-Shyuan Lee, Jamie Yu-Hsuan Chen, Hou-Chuan Lai.

**Software:** Meei-Shyuan Lee.

**Supervision:** Hou-Chuan Lai.

**Validation:** Wei-Chieh Chiu, Zhi-Fu Wu, Yi-Hsuan Huang, Wei-Cheng Tseng.

**Visualization:** Meei-Shyuan Lee.

**Writing – original draft:** Wei-Chieh Chiu, Zhi-Fu Wu, Meei-Shyuan Lee, Yi-Hsuan Huang, Wei-Cheng Tseng.

**Writing – review & editing:** Hou-Chuan Lai.

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
