## [Decision Letter · Decision Letter 0]

17 Oct 2023

PONE-D-23-25558Propofol-based total intravenous anesthesia is associated with less postoperative recurrence than desflurane anesthesia in thyroid cancer surgeryPLOS ONE

Dear Dr. Lai,

Thank you for submitting your manuscript to PLOS ONE. After careful consideration, we feel that it has merit but does not fully meet PLOS ONE’s publication criteria as it currently stands. Therefore, we invite you to submit a revised version of the manuscript that addresses the points raised during the review process.

We look forward to receiving your revised manuscript.

Kind regards,

Stefano Turi

Academic Editor

PLOS ONE

Reviewers' comments:

Reviewer's Responses to Questions

**Comments to the Author**

1. Is the manuscript technically sound, and do the data support the conclusions?

Reviewer #1: Yes

Reviewer #2: Partly

Reviewer #3: Yes

2. Has the statistical analysis been performed appropriately and rigorously? 

Reviewer #1: Yes

Reviewer #2: Yes

Reviewer #3: Yes

3. Have the authors made all data underlying the findings in their manuscript fully available?

Reviewer #1: Yes

Reviewer #2: No

Reviewer #3: Yes

4. Is the manuscript presented in an intelligible fashion and written in standard English?

Reviewer #1: Yes

Reviewer #2: Yes

Reviewer #3: Yes

5. Review Comments to the Author

Reviewer #1: This article analyses the effects of different forms of anaesthesia on postoperative recovery from thyroid cancer surgery based on electronic database and medical records of the TSGH. A series of articles based on data from Taiwan have been previously published in PLOS ONE. The overall structure of this article is similar to other articles and the design of the study is well done. I have no major comments. However the following points need to be revised: 1. The low mortality rate in this study, limited by the characteristics of thyroid cancer, may lead to lower statistical efficacy and needs to be mentioned in the limitations; 2. The addition of some sensitivity analyses could help to enhance the credibility of this study.

Reviewer #2: In this manuscript, the authors conducted the retrospective cohort study observing the effects of desflurane or propofol-based intravenous anesthesia on postoperative recurrence in thyroid cancer surgery.However, there’s still some major concerns that may preclude its publication in the current form.

1) STROBE Statement. Considering as a cohort study, it is better for authors to check all the necessary elements according to the STROBE checklist provided guidance on how to report observational research well, which could achieved on https://www.strobe-statement.org/.

2) Eligible criteria. Why were the patients with ASA score of I not included in the inclusion criteria? It seems to be unusual that all patients with an average age of 45 years undergoing thyroid surgery within ten years had comorbidities preoperatively. It is better to explain why the patients with ASA I were excluded. Furthermore, other factors which may affect immune function should be considered before enrolling cases, such as the use of nerve block, hormone and dexmedetomidine, since they may also interfere with postoperative recurrence by affecting the immune response.

3) Surgical method. Surgical method is the most direct factor for postoperative recurrence. Subtotal thyroidectomy is one of the PTC recurrence risk factors (PMID: 25337182). Incomplete tumor removal is also one of the risk factors (PMID: 34102860). However, in this study, this key factor seems to be missed. Moreover, the condition of secondary surgery during the study period should be considered.

4) Other risk factors for recurrence. Without postoperative 131-I treatment may also be one of the PTC recurrence risk factors (PMID: 25337182). In addition, poor dedifferentiation of the primary tumor is a predictive factor of PTC recurrence (PMID: 23790258). It is better to consider these factors when matching.

5) Follow-up for postoperative recurrence. The authors did not specify how they collected the data of the recurrence.

Reviewer #3: The objective of this paper is very straightforward and easy to understand. The statistical analysis also appears to be generally appropriate. However, there are several points that I would like you to consider:

The authors have applied Cox proportional hazards regression analysis for recurrence events. However, given that death events could be a competing risk, wouldn't it be more appropriate to apply the Fine-Gray hazard regression model?

It seems that the authors did not include variables like surgeon volume, calendar period, and functional status in the propensity score estimation. When estimating the propensity score model, factors that strongly influence the outcome should be prioritized, and collinearity between covariates is generally not a major issue. Therefore, it would be desirable to include these variables in the propensity score model.

For Table 1, it appears that the authors have not calculated the Absolute Standardized Mean Difference (ASMD) for categorical variables. However, Austin & Stuart 2015 (DOI: 10.1002/sim.6607) provides a method for calculating ASMD for categorical variables, which you may find useful. Also, p-values, which are dependent on sample size, are not sufficient evidence to indicate the magnitude of differences between groups, so they could be omitted.

Table 2 presents the results evaluating the association between each covariate and the outcome. However, since the primary objective of this paper is to evaluate the association between the type of anesthesia and the outcome, I do not see significant relevance in this examination.

Table 3 only presents the results showing the relationship between the type of anesthesia and recurrence. Even if there was no significance, it is necessary to also include the intergroup differences in mortality.

For Figure 2, the Kaplan-Meier plot, it is necessary to specify what kind of test was used to calculate the p-values.

6. PLOS authors have the option to publish the peer review history of their article (what does this mean?). If published, this will include your full peer review and any attached files.

Reviewer #1: No

Reviewer #2: No

Reviewer #3: No

---

## [Author Response · Author response to Decision Letter 0]

25 Nov 2023

Please see the file of response to reviewer comments.

---

## [Editor Report · Decision Letter 1]

7 Dec 2023

Propofol-based total intravenous anesthesia is associated with less postoperative recurrence than desflurane anesthesia in thyroid cancer surgery

PONE-D-23-25558R1

Dear Dr. Ho-Chuan Lai,

We’re pleased to inform you that your manuscript has been judged scientifically suitable for publication and will be formally accepted for publication once it meets all outstanding technical requirements.

Kind regards,

Stefano Turi

Academic Editor

PLOS ONE

---

## [Editor Report · Acceptance letter]

29 Dec 2023

PONE-D-23-25558R1 

PLOS ONE

Dear Dr. Lai, 

I'm pleased to inform you that your manuscript has been deemed suitable for publication in PLOS ONE. Congratulations! Your manuscript is now being handed over to our production team.

Kind regards, 

on behalf of

Dr. Stefano Turi 

Academic Editor

PLOS ONE